# The Effect of Hormonal Treatment on Selected Sperm Quality Parameters and Sex Steroids in Tropical Cyprinid Bala Shark *Balantiocheilos melanopterus*



Peter Podhorec [1,*], Jindřiška Knowles [1], Jakub Vysloužil [2], Sergii Boryshpolets [1], Anatolii Sotnikov [1], Martina Holická [2], Jan Kouřil [1] and Borys Dzyuba [1]

1    South Bohemian Research Center of Aquaculture and Biodiversity of Hydrocenoses, Faculty of Fisheries and Protection of Waters, University of South Bohemia in České Budějovice, Zátiší 728/II, 389 25 Vodňany, Czech Republic; matejkovaj@frov.jcu.cz (J.K.); sboryshpolets@frov.jcu.cz (S.B.); asotnikov@frov.jcu.cz (A.S.); kouril@frov.jcu.cz (J.K.); bdzyuba@frov.jcu.cz (B.D.)

2    Department of Pharmaceutical Technology, Faculty of Pharmacy, Masaryk University, Palackeho trida 1946/1, 612 00 Brno, Czech Republic; jakub.vyslouzil@gmail.com (J.V.); 93holicka.martina@seznam.cz (M.H.)

*    Correspondence: podhorec.peter@seznam.cz





**Simple Summary:** The bala shark *Balantiocheilos melanopterus* does not reproduce naturally under culture conditions, and hormone stimulation is routinely administered to male broodstock to ensure production of adequate quantities of high-quality sperm. Poly(lactic-co-glycolic acid) (PLGA) microparticles with slow release of 10 µg/kg fish body weight GnRHa were injected to induce spermiation in bala sharks and to compare them with standard treatments in tropical cyprinid culture: recombinant hCG and GnRHa with metoclopramide. The PLGA microparticle treatment led to significantly higher plasma T and 11-KT concentrations at 24 h post-injection than was detected in the control group. Sperm motility parameters were evaluated during the 10–60 s after motility activation at 2 s intervals. Starting from 28 s post-activation, the motility percentage was significantly higher in the PLGA group compared to the control group. The PLGA microparticle treatment was also found to significantly increase sperm volume and total sperm count compared to control (0.9% NaCl). The PLGA microparticle system with continuous release of GnRHa was identified as optimal for inducing spermiation in cyprinid bala sharks.

**Abstract:** Hormone treatments are routinely employed with bala shark *Balantiocheilos melanopterus* male broodstock to stimulate the production of high-quality sperm. In the current trial, three spermiation-inducing treatments were evaluated: 20 µg/kg body weight (BW) recombinant hCG; 20 µg/kg BW [D-Ala⁶, Pro⁹, NEt]- gonadotropin-releasing hormone (GnRHa) with 20 mg/kg BW metoclopramide; and poly(lactic-co-glycolic acid) (PLGA) microparticles with slow release of 10 µg/kg BW GnRHa. A 0.9% saline AS negative control was also included. Administration of the GnRHa through the form of slow release of PLGA microparticles 24 h after treatment resulted in a significantly higher sperm volume, motility percentage, and total sperm count compared to the control. Injection of GnRHa with metoclopramide induced sperm parameters that did not differ from the control, with the exception of motility percentage. The lowest potency to induce spermiation in bala sharks was in the treatment with recombinant hCG. Both PLGA microparticles and GnRHa with metoclopramide significantly increased blood plasma concentrations of testosterone and 11-ketotestosterone compared to the control. The PLGA microparticle system with continuous release of 10 µg/kg BW GnRHa was the most effective treatment in inducing spermiation in bala sharks.

**Keywords:** sperm; cyprinid; sustained release; reproduction; luteinizing hormone

## 1. Introduction

The bala shark *Balantiocheilos melanopterus* is an omnivorous and pelagic cyprinid species native to rivers and lakes of Sumatra and Borneo in Southeast Asia [1]. During the rainy season, the bala shark migrates to reach specific breeding grounds where it undergoes mass spawning [2]. It is a popular aquarium species, with natural populations currently declining in much of its native range because of habitat degradation and overfishing for the aquarium trade [3]. It is listed as vulnerable on the IUCN Red List of Threatened Species, with only cultured specimens available for the pet trade.

Bala sharks do not reproduce spontaneously under culture conditions, and hormone therapy is necessary to stimulate final oocyte maturation and ovulation [4]. Although bala shark males usually complete spermatogenesis and spermiation in captivity, the quality of sperm obtained is variable. Hormone treatments are routinely employed to ensure an adequate quantity of high-quality sperm. Hormone treatments can be categorized, according to mode of action, as gonadotropin preparations, e.g., human chorionic gonadotropin (hCG); treatments involving the carp pituitary acting at the level of the gonad [5]; and gonadotropin-releasing hormone analogues (GnRHa) with or without dopamine antagonists, acting on the pituitary and hypothalamus [6]. GnRHa is a small peptide of low molecular weight that can be administered in sustained-release delivery systems [7]. In contrast to the situation in most mammals, extended exposure of fish organisms to exogenous GnRHa stimulation does not cause desensitization of the pituitary gonadotrophs in fish, but elevates levels of plasma luteinizing hormone (LH) and stimulates the natural progression of steroidogenesis associated with spermiation [8].

Common sustained-release systems in aquaculture include ethylene-vinyl acetate copolymer (EVAc) implants [9], solid implantable cholesterol pellets [10], and poly(lactic-co-glycolic acid) (PLGA) microparticles [11]. Poly(lactic-co-glycolic acid) is biodegradable, has excellent biocompatibility, and is safe for use in pharmaceutical products [12]. Encapsulated GnRHa is released via degradation and erosion of the polymer matrix [13], leading to prolonged stimulation of LH [8]. The PLGA microparticles have the advantage that they are administered as a liquid suspension, enabling fish-specific dosing based on body weight. Treatment with GnRHa through sustained-release systems is reported to significantly increase the quantity and quality of expressible milt in several fish species [14–16]. Most trials of sustained-release systems have focused on marine species [8], with the few studies in freshwater aquaculture producing suboptimal results [10,17,18].

An essential prerequisite of successful reproduction in captivity is synchronization of maximum sperm production with ovulation and collection of ripe eggs. This is especially crucial in bala shark, as numbers of available male broodstock are usually low, and the species' low tolerance to manipulation and handling may have fatal consequences (unpublished observation).

The goal of the current study was to assess the efficacy of sustained-release GnRHa from PLGA microparticles compared to the single-dose treatments commonly used in tropical cyprinids (recombinant hCG and GnRHa with dopamine antagonist) in stimulating spermiation in bala sharks. Evaluation of the experimental treatments was based on the qualitative and quantitative characteristics of the obtained sperm and plasma sex steroid profiles.

## 2. Materials and Methods

### 2.1. Animals and Maintenance

Subadult bala shark broodstock (two years old) were purchased in 2018 from a local ornamental fish importer and held for one year in an aquaculture system comprising separate 600 L tanks (dimensions: 120 cm × 100 cm × 50 cm) with mechanical and biological filtration systems at the University of South Bohemia in Ceske Budejovice. The photoperiod was set to 12:12 L:D. Tap water (after dechlorination by activated carbon) with the following characteristics was used (mean ± SD): pH 8.2 ± 1.9, electrical conductivity 290 ± 35 µS/cm at 25 °C, temperature 26.0 ± 0.5 °C, and dissolved oxygen 7.8 ± 0.4 mg/L. Every seven days, 50% of the water was changed. Fish were fed twice daily to apparent

satiation using commercial extruded feed for ornamental fish (Tetra Discus; Tetra GmbH, Melle, Germany), with a daily addition of frozen bloodworms. After one year, males averaging $180 \pm 40$ g body weight (BW) were randomly divided into four groups of 10 and experimental treatments were applied. No significant differences in BW were found among experimental groups using one-way ANOVA ($p < 0.05$).

### 2.2. Hormone Treatments

2.2.1. Preparation of PLGA Microparticles with Continuous GnRHa Release

The microparticles were prepared by solvent evaporation from a multiple water/oil/water emulsion. During evaporation, solid spherical particles are formed from the polymer, which entraps the drug in its structure. Alarelin (APExBIO, Houston, TX, USA), a GnRH analogue, was used as a hormonally active substance. A copolymer of 75% polylactic acid and 25% polyglycolic acid (PLGA) was used as carrier material (Resomer 753, Evonik, Darmstadt, Germany).

For the oil phase, 800 mg of PLGA of the appropriate Resomer and 5 g of dichloromethane (Penta, Prague, Czech Republic) were weighed into a tube and allowed to dissolve. The inner aqueous phase was prepared by dissolving gelatin in purified water at 65 °C (10%) (Sigma Aldrich, St. Louis, MO, USA). The external aqueous phase consisted of two solutions: a premixed 1% polyvinyl alcohol (PVA) (Sigma Aldrich, St. Louis, MO, USA) and the main continuous aqueous phase with 0.1% PVA.

The next step consisted of preparation and further processing of the emulsion from the individual phases. On an analytical balance, 10 mg of GnRHa was weighed into a microcentrifuge tube, and 1.5 mL of a 10% gelatin solution was added using a syringe. The microcentrifuge tube with prepared internal aqueous phase was vortexed to dissolve the drug. The resulting solution was then poured into the wide-necked tube containing the PLGA/dichloromethane oil phase (800 mg/5 g) and vortexed for 30 s to ensure primary emulsification through the formation of a crude primary water/oil emulsion. The emulsion was then mixed for 1 min using a homogenizer (T25 basic, IKA-Werke, Staufen, Germany), converting the dispersed phase to smaller droplet sizes and producing a fine emulsion. Subsequently, 12 g of a 1% PVA solution was added to a wide-necked tube and homogenized for 1 min to produce a concentrated multiple water1/oil/water2 emulsion. The emulsion was then added to a larger beaker with 200 mL of a 0.1% PVA solution with 2% NaCl for dilution, and the contents were stirred for two hours using a mechanical stirrer at 450 rpm. During this time, the organic solvent evaporated, and the polymer solidified into spherical particles. The resulting micro-suspension was filtered through a 250 μm screen for the eventual separation of agglomerates. Isolation of the microparticles was performed by continuous centrifugation ($3461 \times g$ for 2 min). Excess water was decanted, and the microparticles were collected in Petri dishes, stored in a freezer, and dried by lyophilization.

The alarelin content in PLGA microparticles was determined using HPLC. The PLGA microparticles were dissolved in acetone. The resulting solution was mixed 1:1 ($v/v$) with a phosphate buffer of pH 7.0 (Fisher Scientific, spol. s.r.o., Pardubice, Czech Republic). The mixture was filtered through a 0.45 μm membrane filter. The content was quantified by HPLC (Agilent 1100, Agilent, Santa Clara, CA, USA) with a NUCLEODUR 100-5 CN-RP (150 mm $\times$ 4.6 mm, 5 μm) column. A binary mixture of acetonitrile and 20 mM $H_3PO_4$ (16:84, $v/v$) was used as a mobile phase with a flow rate of 0.8 mL/min, with the temperature set at 30 °C, an injected sample volume of 20 μL, and a 220 nm detection wavelength.

For the drug-release study, 50 mg of microparticles was suspended in 1%/0.4 mL agarose solution in a glass vial and cooled to solidify the agarose, then 800 μL agarose was added as covering layer. When the agar had solidified, 5 mL phosphate buffer was added. Precisely 2 mL of the buffer was collected after 4, 24, 48, 72, 96, and 168 h and filtered through a 0.22 μm membrane filter.

After each sampling, the buffer residue in vials was removed, the vials were washed with 0.5 mL buffer, and 5 mL of fresh buffer was added. In vitro experiments were run at

5 °C and performed in triplicate for each sample. Collected samples were analyzed using the HPLC method as described.

Prepared PLGA microparticles contained 45 µg of Alarelin per 100 mg of sample (encapsulation efficiency of 43%). The release kinetics in Figure 1 show, that, over the course of seven days, approximately one-third of the drug was released (144.75 µg per 100 mg of the sample). However, the main portion of the release took place during first 72 h. During this time, the drug was released with almost regular increments per 24 h (51.1 µg/24 h; 90.90 µg/48 h; 123.31 µg/72 h), very similar to advantageous zero-order kinetics. The sample was treated as a delivery system with 1.2 µg of Alarelin released/mg of PLGA microparticles/72 h.

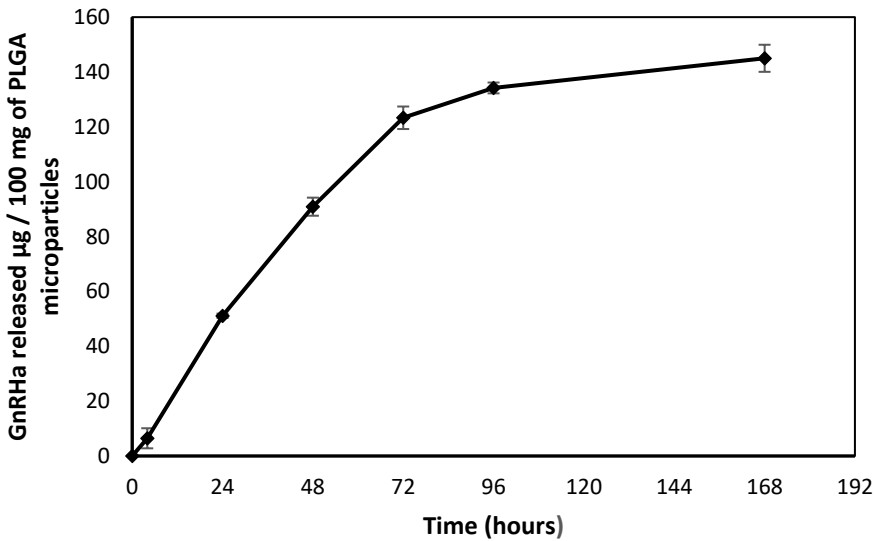

**Figure 1.** Kinetics of release of GnRHa from PLGA microparticles.

### 2.2.2. Hormone Treatments

To evaluate the effectiveness of PLGA microparticles with GnRHa in stimulating bala shark sperm release, comparisons were made with both negative (0.9% NaCl) and positive controls. As positive control groups, commonly used treatments based on GnRHa with dopamine antagonist and recombinant hCG were selected. The doses were established based on previous successful trials with cyprinids for combined treatment [19–21] and for hCG treatment [22,23]. In our recently published study, we have shown excellent tolerance in carp to the PLGA microparticle doses used in current study [24].

Male fish were administered a single intramuscular injection of:

(1)   1 mL/kg BW 0.9% NaCl (Braun Melsungen AG, Melsungen, Germany); control group.
(2)   20 µg/kg BW recombinant hCG (Ovitrelle, Merck Europe B.V., Amsterdam, The Netherland); hCG group.
(3)   25 µg/kg BW [D-Ala⁶, Pro⁹, NEt]-GnRH (APExBIO, Houston, TX, USA) combined with 20 mg/kg BW metoclopramide (Met) (Sigma-Aldrich, St. Louis, MO, USA); GnRHa + Met group.
(4)   10 µg/kg BW [D-Ala⁶, Pro⁹, NEt]-GnRH in PLGA microparticles; PLGA group.

All substances were dissolved in 0.9% NaCl.

### 2.3. Collection and Analysis of Samples

#### 2.3.1. Blood and Sperm Sample Collection

Heparinized needles in 1 mL syringes were used to collect blood samples (400 µL) by caudal venipuncture before injection and 24 h post-injection (PI). Blood samples were centrifuged at $4000 \times g$ for 10 min at 8 °C, and plasma was stored at −80 °C until analysis. Fish were anaesthetized with 0.03 mL/L clove oil before manipulation.

Sperm was collected once at 24 h post-treatment, as attempts at multiple sperm collection led to health problems and mortality. The urogenital pore was dried with a paper towel before direct sperm collection. To counteract urine contamination, sperm was collected into 5 mL syringes containing 2.0 mL of Kurokura 180 immobilizing solution [25]. After measuring total sperm volume, syringes were immediately placed on ice and transported to the laboratory for analysis. Sperm samples from individual males were stored on ice at 4 °C for not longer than two hours during motility analysis.

### 2.3.2. Testosterone and 11-Ketotestosterone Analysis

Plasma levels of testosterone (T; KAPD1559; DIAsource ImmunoAssays SA, Louvain-la-Neuve, Belgium) and 11-ketotestosterone (11-KT; 582751; Cayman Chemical, MI, USA) were evaluated by ELISA using commercially available kits according to the manufacturer's instructions, with each standard and plasma sample run in duplicate. The intra-assay coefficients of variation for T and 11-KT were less than 6% in all tests, and inter-assay coefficients of variation were less than 7% for T and 11-KT. The absorbance of all assays was read by a PlateReader AF2200 microplate reader (Eppendorf Czech and Slovakia s.r.o., Říčany u Prahy, Czech Republic).

### 2.3.3. Sperm Production Indexes

The volume of sperm was estimated by measurement of sperm sample mass to the nearest 10 mg. The sperm concentration of each sample was estimated using a Burker cell hemocytometer (Meopta, Prerov, Czech Republic) at $200\times$ magnification on an Olympus BX 50 phase-contrast microscope (Olympus Czech Group, Prague, Czech Republic). Total sperm count was computed as sperm concentration multiplied by the volume of the sperm sample. Normalized-by-male-BW sperm volume (mL/kg) and sperm count (TSP, $10^{10}$ spz/kg) were presented as sperm production indexes.

### 2.3.4. Sperm Motility Analysis

Sperm was activated in distilled water containing 0.125% Pluronic F-127 (catalogue number P2443, Sigma-Aldrich) to avoid sperm sticking to the glass slide. Motility records were made from the bottom part of the drop. Motility was recorded at 50 fps using optical negative phase-contrast microscopy, a $\times10$ magnification lens (PROISER, Madrid, Spain), and an IDS digital camera (IDS Imaging Development Systems GmbH, Obersulm, Germany). The total number of spermatozoa in which motility parameters were analyzed at each time point ranged from 910 to 2774. Altogether, 198,009 spermatozoa were analyzed. The videos were recorded for the first 60 s after motility activation, and kinetic data of sperm motility were collected at 2 s intervals beginning at 10 s post-activation using the CASA plugin for ImageJ (Purchase, Earle, 2012). Kinetic parameters obtained by CASA for all sperm samples used in the study were subjected to a correlation analysis using Spearman's rank correlation coefficient. To simply data presentation, only parameters with a low correlation coefficient (r < 0.06) were selected as descriptors of sperm motility. These parameters were the percentage of motile cells, curvilinear velocity (VCL) in μm/s and the linearity of track (LIN).

### 2.4. Statistical Analysis

### 2.4.1. Kinetic Parameter Analysis

Kinetic parameters were subjected to a correlation analysis using Spearman's rank correlation coefficient. All spermatozoa with VCL < 10 μm/s were considered immotile and excluded from the analysis. The percentage of motile spermatozoa and VCL and LIN values for each combination of male/experimental group/time post-activation were extracted from the CASA dataset. The mean motility rate for each male was used to plot trend lines for the motility rate at 10–60 s post-activation. Quadratic polynomial regression was selected for visualizing motility trends. These trend lines were used to determine the "time points of interest" at which the significance of differences among groups was additionally evaluated. Before analysis, the data were tested for normality

and homogeneity of variance using Kolmogorov–Smirnov and Levene's tests, respectively. All studied parameters were normally distributed and had similar dispersion values; the data were first analyzed by two-way ANOVA. Factor "treatment" (four levels: control, hCG, GnRHa + Met, and PLGA) and "time" (25 time post-activation time points) were significant for VCL, LIN, and motility percentage at $p < 0.001$, and interaction of factors was insignificant for VCL and LIN (at $p > 0.1$) and significant for motility percent ($p < 0.01$). Tukey's test was used to quantify differences among treatments (mean values of the motility rate, VCL, and LIN in individual males) at each sampling time and between time points of interest for each experimental treatment.

#### 2.4.2. Total Sperm Count

As data for the TSP were normally distributed and showed no significant differences in dispersion values (Kolmogorov–Smirnov and Levene's tests, respectively), parametric one-way ANOVA was applied, and Tukey's honest significant difference test was used to assess differences among groups.

#### 2.4.3. Testosterone and 11-Ketotestosterone Analysis

Sex steroid values were not normally distributed and showed significant differences in dispersion values (Kolmogorov–Smirnov and Levene's tests, respectively, $p < 0.05$). The Kruskal–Wallis test was used to analyze differences among groups, followed by multiple comparisons of mean ranks for all groups. These tests were applied separately to compare the treatment groups at different times PI and at the same time PI.

Statistical analysis and graph plotting were performed using Statistica v. 13.5.0.17 (TIBCO Software Inc., Palo Alto, CA, USA). Null hypotheses were rejected at $p < 0.05$ in all applied statistical tests.

### 3. Results

#### 3.1. Plasma Concentration of Testosterone and 11-Ketotestosterone

The PLGA sustained-release system and GnRHa + Met treatment led to significantly higher plasma T concentrations at 24 h PI than detected in the control group. Treatment with hCG did not elicit plasma T concentrations significantly different from the control group ($p < 0.05$) or other experimental groups (Figure 2a).

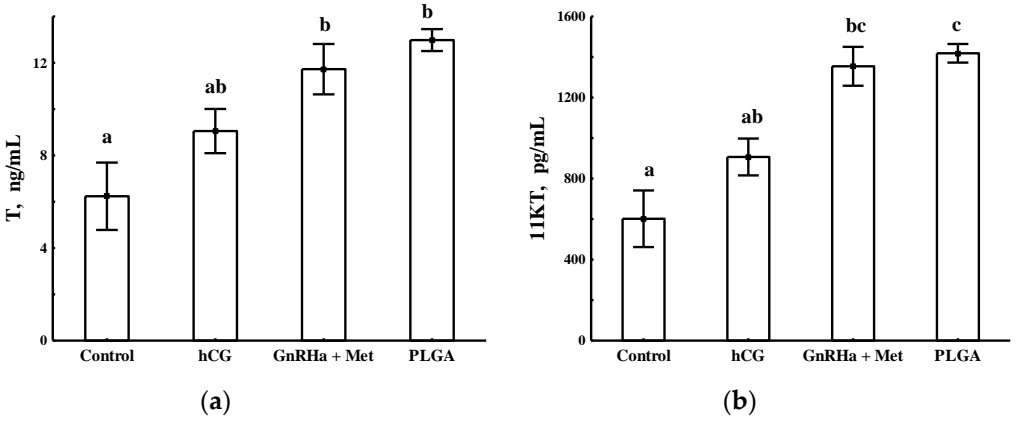

(a)　　　　　　　　　　　　　　(b)

**Figure 2.** (**a**) Plasma testosterone (T) concentrations in bala sharks after hormone treatment; (**b**) plasma 11-ketotestosterone (11-KT) concentration after hormone treatment in bala sharks. Control (1 mL/kg 0.9% NaCl); hCG (20 µg/kg recombinant hCG); GnRHa + Met (25 µg/kg [D-Ala[6], Pro[9], NEt]-GnRH with 20 mg/kg metoclopramide); PLGA (PLGA microparticles). Different letters indicate significant differences ($p < 0.05$; Kruskal–Wallis test, multiple comparisons of mean ranks for all groups). Data are expressed as the mean $\pm$ SE.

The PLGA sustained-release system and GnRHa + Met treatment led to significantly higher plasma 11-KT concentrations at 24 h PI ($p < 0.05$) than detected in the control group. Treatment with hCG did not elicit plasma 11-KT concentrations significantly different from the control group ($p < 0.05$) or GnRHa + Met. Significantly lower 11-KT values were found after hCG treatment compared to the PLGA at 24 h PI (Figure 2b).

### 3.2. Sperm Production Indexes

All males in the trial produced viable sperm.

The PLGA group showed significantly higher normalized sperm volumes than the control and hCG group but did not differ from the group treated with GnRHa + Met. No differences were found among control, hCG, and GnRHa + Met groups in sperm volume ($p < 0.05$) (Figure 3a). No significant differences in sperm concentration were detected among experimental groups (Figure 3b).

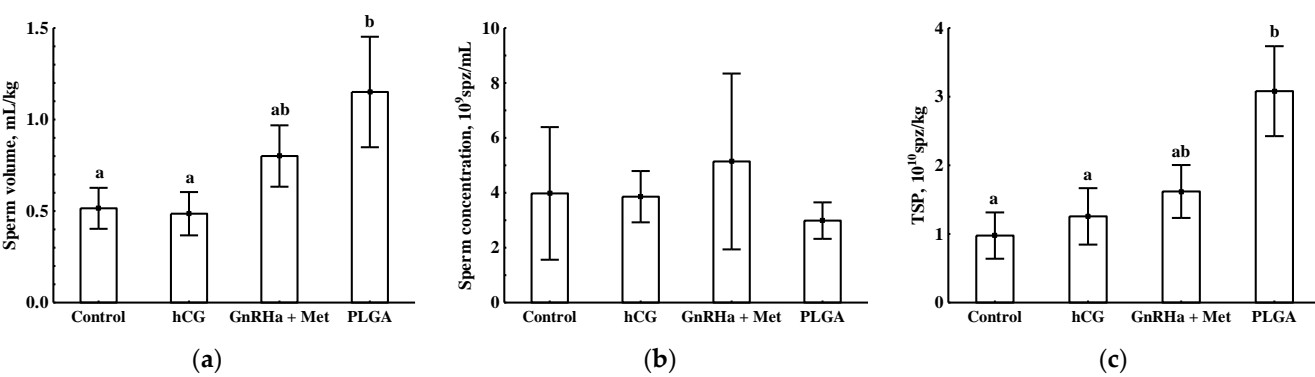

**Figure 3.** Sperm production indexes after hormonal treatment in bala sharks. (**a**) Normalized sperm volume; (**b**) sperm concentration; (**c**) normalized total sperm count after hormonal treatment in bala sharks. Control (1 mL/kg 0.9% NaCl); hCG (20 µg/kg recombinant hCG); GnRHa + Met (25 µg/kg [D-Ala$^6$, Pro$^9$, NEt]-GnRH with 20 mg/kg metoclopramide); PLGA (PLGA microparticles). Different letters indicate significant differences ($p < 0.05$; one-way ANOVA, Tukey's HSD test). Data are expressed as the mean ± SE.

The PLGA group showed significantly higher total sperm counts than the control group and the group treated with hCG but did not differ from the group treated with GnRHa + Met. No differences were found among control, hCG, and GnRHa + Met groups in total sperm counts ($p < 0.05$) (Figure 3c).

### 3.3. Sperm Motility Parameters

After sperm motility activation VCL, LIN, and motility percentage were dynamically changed in a treatment-specific way, and the obtained regression lines made it possible to determine the "points of interest" for the next step of the statistical analysis (Figure 4). Generally, the motility percentage and VCL decreased 18–20 s post-activation in all experimental groups.

However, at the initial stage of motility (10 s post-activation), motility percentage was significantly lower in the GnRHa + Met than in all other groups, while no significant differences in average VCL among experimental groups at this post-activation time point were found. Starting from 30 s post-activation, motility percentage was significantly higher in the PLGA group compared to the control group, and starting from 34 s post-activation, it was also higher in the PLGA group compared to both control and hCG groups, and these dependencies were the same until 60 s post-activation. A significant increase in motility percentage between 10 s and 22 s post-activation was observed in the GnRHa + Met group only, and no significant rise in VCL was found between 10 s and 20 s post-activation in all groups. A significant decrease in VCL (in comparison to 10 s post-activation) was observed after 20 s post-activation in the GnRHa + Met group and 28 s post-activation in the control

and hCG groups. Significantly lower LIN in the PLGA group in comparison to the hCG group was found at late stages of motility (38–60 s post-activation).

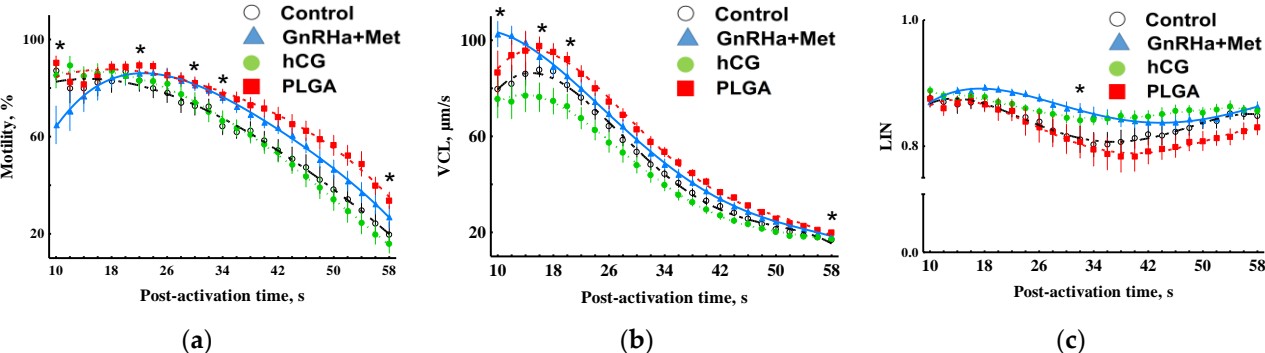

**Figure 4.** Dynamics of motility parameters during post-activation time for Bala shark spermatozoa obtained after hormonal treatments. (**a**) Sperm motility percentage (Motility); (**b**) curvilinear velocity (VCL); (**c**) sperm track linearity (LIN). Control (1 mL/kg 0.9% NaCl); hCG (20 µg/kg recombinant hCG); GnRHa + Met (25 µg/kg [D-Ala$^6$, Pro$^9$, NEt]-GnRH with 20 mg/kg metoclopramide); PLGA (PLGA microparticles). Data are presented as means (dots) $\pm$ SE (whiskers), lines are quartic polynomial regression lines, * indicates a "point of interest" discussed in the text.

## 4. Discussion

Injection of PLGA microparticles with continuous GnRHa release stimulated a significant increase in sperm volume and total sperm count, but not motility parameters, at the initial stage of motility in the sperm of bala shark males compared to a control group.

Similar results have been reported in marine fish species after administration of sustained-release systems, showing increased quantity [14–16] and quality [26] of expressible sperm. The use of sustained-release delivery systems in marine aquaculture is based on the need for prolonged stimulation of LH levels in species with asynchronous oocyte development and multiple spawning [8]. Administration of a single GnRHa injection is insufficient due to its short residence time in circulation [27]. On the other hand, in most commercially important freshwater cyprinids, a single injection of GnRHa with a dopamine antagonist is an effective inducer of an LH surge, leading to enhanced spermiation [19], and, in the trials performed so far, significantly outperforming sustained-release systems [17,18]. However, our data show that 24 h sustained release of GnRHa alone led to significantly higher sperm volumes, motility rates, and total sperm counts compared to untreated controls. In contrast, treatment with combined GnRHa and dopamine antagonist did not enhance sperm compared to the control group except for the motility rate.

The observed increase in bala shark sperm volume 24 h after administration of the sustained release system can be attributed to a stimulatory effect on seminal fluid production, which leads to hydration of testes [28] and acquisition of motility capacity in the existing intra-testicular sperm [5]. The increased fluid content of the testes allowed stripping of more sperm, which are present within the testes but otherwise would not be released. No decrease in sperm concentration was found after injection with the sustained-release system, despite significantly increased sperm volume compared to the control group.

Hormone induction does not usually influence sperm quality parameters, such as motility rate, motility duration, or sperm velocity, in captive species with moderate spermiation ability [16]. Improvement of sperm quality parameters is more characteristic of species with severe reproductive dysfunctions and little or no spermiation under captive conditions, such as sturgeon [18] and flatfishes [26]. All bala shark males in our trial completed spermiogenesis, and expressible milt could be obtained prior to treatment. In addition, as minor differences among the experimental groups in comparison to control were found in terms of motility percentage, VCL, and LIN, it can be summarized that hormonal treatment by PLGA led to production of greater amounts of sperm that could be collected but not to increased sperm quality parameters. In studies of freshwater and marine fishes [4,18,19],

significantly increased seminal plasma and sex steroid values linked to higher motility rates have been reported. In contrast to the improvement in sperm motility, no effect on the VCL or LIN was noticed when using the PLGA system and GnRHa + Met.

The least effective of the tested treatments was the recombinant hCG at 20 μg kg$^{-1}$ BW, approximately equal to 500 I.U., which produced results similar to the control group. A possible explanation could be the lower affinity of gonadotropin receptors for the mammalian recombinant protein, as has been suggested for other cyprinids [29,30].

The PLGA microparticles with continuous GnRHa release and GnRHa + Met both significantly increased blood plasma concentration of T and 11-KT compared to the control group. Androgens 11-KT and T are primary sex steroids responsible for the initiation and progression of spermatogenesis [31]. 11-KT stimulates the development of secondary sexual characteristics, spermatogonial proliferation, and spermiation [32], whereas T, a biosynthetic precursor of 11-KT, stimulates spermatogenesis [33]. Increased levels of 11-KT and T provide evidence of their roles in facilitating spermiation in bala sharks and underscore the positive effect of the PLGA microparticle system with continuous GnRHa release on the quantity and quality of bala shark sperm.

Release of GnRHa from PLGA microparticles occurs via diffusion and homogeneous bulk erosion of the biopolymer [13] and is characterized by an initial burst immediately after administration and a sustained or continuously declining release until depletion of the microspheres (Figure 1). We suggest that the high initial burst of GnRHa with subsequent decline might be the reason for the potent effect versus other treatments, as well as compared to the results of studies with other freshwater species that used systems with more gradual GnRHa release [17,18]. It is unclear whether constantly elevated plasma LH in treatments with GnRHa delivery systems reflects the natural physiologic situation necessary for gonadal steroidogenesis changes [5]; nevertheless, it seems to induce the appropriate hormonal changes for triggering gonad maturation and the production of viable sperm.

During preparatory work for this study, contamination of bala shark sperm by urine was observed, leading to premature sperm activation. To ensure the high fertilizing ability of Bala shark sperm, we strongly recommend the collection of sperm into immobilizing solution, e.g., Kurokura 180 [25], to counteract this problem.

## 5. Conclusions

Based on our data, we can conclude that a PLGA microparticle system (75% polylactic acid; 25% polyglycolic) with continuous release of 10 μg kg$^{-1}$ of GnRHa in bala sharks is a potent inducer of sperm of both high quality and quantity and, together with the possibility of precise fish-specific dosing, represents an effective means of sperm induction. Another advantage of the PLGA delivery system with continuous GnRHa release in artificial reproduction of cyprinids might be the elimination of the so-far necessary addition of dopamine antagonists in cyprinid reproductive techniques. However, more research on sustained-release systems in freshwater fish reproductive techniques is needed.

**Author Contributions:** Conceptualization, P.P.; methodology, P.P., B.D., S.B., A.S., J.V., M.H.; software, S.B.; validation, P.P., B.D., J.V.; formal analysis, P.P.; investigation, P.P., J.K. (Jindřiška Knowles), J.K. (Jan Kouřil); resources, P.P.; data curation, B.D.; writing—original draft preparation, P.P., B.D.; writing—review and editing, B.D.; visualization, B.D.; supervision, P.P., B.D.; project administration, P.P.; funding acquisition, P.P. All authors have read and agreed to the published version of the manuscript.

**Funding:** This research was funded by the Ministry of Agriculture of the Czech Republic, grant number NAZV QK1810221.

**Institutional Review Board Statement:** The study was conducted according to the guidelines of the Declaration of Helsinki and followed national and international guidelines for the protection of animal welfare (EU-harmonized Animal Welfare Act of the Czech Republic). The experimental unit is licensed with no. 2293/2015-MZE-17214 and no. 55187/2016-MZE-17214 within the project

NAZV QK1810221 according to the Czech National Directive (the Law against Animal Cruelty, no. 246/1992).

**Data Availability Statement:** The data presented in this study are available on request from the corresponding author.

**Acknowledgments:** The authors give special thanks to Ing. Pavel Šablatura for his technical assistance.

**Conflicts of Interest:** The authors declare no conflict of interest.

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
