# Peer review of "The Effect of Hormonal Treatment on Selected Sperm Quality Parameters and Sex Steroids in Tropical Cyprinid Bala Shark Balantiocheilos melanopterus"

_fishes, doi:10.3390/fishes7030122_

Round 1

Reviewer 1 Report

The study compares the use of different hormonal systems to induce final sperm maturation in cyprinid bala shark Balantiocheilos melanopterus, evidencing good results using a slow delivery system of GnRHa.

L16 were injected

L17 and to compare with

L20 28 s

L20 starting from the….When the sperm was evaluated 28 s after the activation…?

L31 How long after the treatment?

L45 extirpated?...alternative Word?

L45, 74, 167, 360 Check the number size

L81 commonly used

L100 Sex ratio? All males?

L114 Define PVA

L120, 221, 222, 223, 232, 236 sec > s

L139 20 mM

L142, 144, 148 was > were?

L152 43%

L156(x3), 158, 194, 342 hrs > h

Fig 1 (axis). 100 mg of

L164 Move “were selected” to the nd of the sentence.

Table 1 could be deleted because is repeating information already shown in L170-179

L188 at/of?

L217 frames/sec > fps

L219 Delete ).

L250 Avoid bold letters

References. Use italics to write species names

Author Response

Everything was corrected according recviewers suggestions.

L16 were injected

L17 and to compare with

L20 28 s

L20 starting from the….When the sperm was evaluated 28 s after the activation…?

L31 How long after the treatment?

L45 extirpated?...alternative Word?

L45, 74, 167, 360 Check the number size

L81 commonly used

L100 Sex ratio? All males? All males

L114 Define PVA - it is defined in the first part of the sentence

L120, 221, 222, 223, 232, 236 sec > s

L139 20 mM

L142, 144, 148 was > were?

L152 43%

L156(x3), 158, 194, 342 hrs > h

Fig 1 (axis). 100 mg of

L164 Move “were selected” to the nd of the sentence.

Table 1 could be deleted because is repeating information already shown in L170-179 – was deleted

L217 frames/sec > fps

L219 Delete ).

L250 Avoid bold letters

Reviewer 2 Report

The study describes the utilization of Poly(lactic-co-glycolic acid) (PLGA) in administering GnRHa to stimulate spermiation of the bala shark males. The study is very well conceptualized and well designed. All parts of the manuscript are well written and have a logical sequence and progression. I believe the study has significant scientific merit and will be of interest for the readership of the ‘Fishes’ journal. With that said, I have only few minor comments that should be addressed before publishing.

Line 14: hormonal stimulation, not therapy. Please correct it throughout the text

Line 22: spermatozoa

Line 30: 0.9% saline AS negative control

Line 47: The reference number is in superscript. Please correct this throughout the manuscript

Line 136: Please write the composition of the phosphate buffer

Line 214: Why was the sperm activated in the distilled water? Doesn’t that pose too much of an osmotic shock? Especially since plenty of activation solutions for cyprinid species already exist in the literature.

Section 2.3.4., and the rest of the manuscript: Write ‘seconds’ uniformly as either ‘sec’ or ‘s’ throughout the text. Also, the unit should be written separately from the number.

Line 219: Double ‘).’

Line 250: TSP in bold

Line 279: The analysis conducted could be written here, similarly to Fig. 3

Line 289: The fact that there is no significant difference between PLGA and GnRHa + Met groups in the normalized spermatozoa count (Fig. 3c) is very strange when looking at the Figure. Even if its so, I would emphasize more that the PLGA group did yield much higher average. Maybe insert a sentence saying ‘Even though there were no statistical differences, the PLGA group showed much higher average …’, or something to that effect.

Line 298: I do not understand why do authors constantly insist on writing ‘spermatozoon’ in the manuscript. This is absolutely and completely wrong!!! Either write ‘spermatozoa’ because you are evaluating the parameters for all cells in the milt sample, i.e. plural, or omit mention this word. This needs to be corrected throughout the manuscript!!!

Line 305: double space in the middle of the row

Author Response

All corrections were accepted and manuscript was corrected accordingly

Line 14: hormonal stimulation, not therapy. Please correct it throughout the text

Line 22: spermatozoa

Line 30: 0.9% saline AS negative control

Line 47: The reference number is in superscript. Please correct this throughout the manuscript

Line 136: Phosphate buffer – Fishers Scientific, Pardubice, Czech Republic

Line 214: Why was the sperm activated in the distilled water? Doesn’t that pose too much of an osmotic shock? Especially since plenty of activation solutions for cyprinid species already exist in the literature.

Sperm was activated in distilled water containing 0.125% Pluronic. We selected this activation condition based on our initial trials, which showed that sperm motility in this condition is long enough for analysis, which we performed. You are right; there are several activating media described for carps. However, they were designed to improve sperm motility and prolong sperm motility duration. For our study, this point was unnecessary, and we decided to use this medium precisely to avoid considering the influence of the ionic composition of activating medium on sperm motility.

Section 2.3.4., and the rest of the manuscript: Write ‘seconds’ uniformly as either ‘sec’ or ‘s’ throughout the text. Also, the unit should be written separately from the number.

Line 219: Double ‘).’

Line 250: TSP in bold

Line 279: The analysis conducted could be written here, similarly to Fig. 3

Line 289: The fact that there is no significant difference between PLGA and GnRHa + Met groups in the normalized spermatozoa count (Fig. 3c) is very strange when looking at the Figure. Even if its so, I would emphasize more that the PLGA group did yield much higher average. Maybe insert a sentence saying ‘Even though there were no statistical differences, the PLGA group showed much higher average …’, or something to that effect.

Yes, the difference was not significant (p=0.14). According to your suggestions, the sentence “However, even though there were no statistical differences (p = 0.14), the PLGA group showed much higher average TSP compared to GnRHa+Met group” is added.

Line 298: I do not understand why do authors constantly insist on writing ‘spermatozoon’ in the manuscript. This is absolutely and completely wrong!!! Either write ‘spermatozoa’ because you are evaluating the parameters for all cells in the milt sample, i.e. plural, or omit mention this word. This needs to be corrected throughout the manuscript!!!

That appeared since “A noun working as an adjective can never be plural”. We agree that in modern scientific literature, mentioned combinations are not frequently used, and we follow your recommendations. 

Line 305: double space in the middle of the row

This manuscript is a resubmission of an earlier submission. The following is a list of the peer review reports and author responses from that submission.

Round 1

Reviewer 1 Report

As per attached edited PDF

Author Response

All recommendations of the reviewer were accepred and manuscript was changed accordingly.

Reviewer 2 Report

The paper is aimed on the hormone induction of spermiation in bala shark. Authors set out relatively difficult analyses for this species of fish. This increases the scientific value of the research. Nevertheless, I have following questions, comments, and remarks about the presented work:

I do not fully identify with the title of the publication. Although the authors monitored some fish spermatozoa quality parameters and hormone production, other complex markers are missing. Spermiation involves a broader complex of observations (for example morphology; ectoplasmic specialization; initiation, progression, and regulation of spermiation etc.) in addition to the monitored parameters. Interactions of various structures, cellular processes and other changes are missing.

Based on the above, I recommend editing the title: e.g. “The effect of hormonal treatment on selected sperm quality parameters and sex steroids in tropical cyprinid bala shark.”

  1. Introduction:

The introduction meets the criteria for this type of study, and I have no comments on this section.

  1. Material and Methods:

In this section I have following comments, recommendations, or questions:

Page 2 Line 88 - I recommend adding dimensions of the water tank (the volume itself may not be the authoritative parameter).

In this section, I recommend adding a list of chemicals to increase the clarity of the selected procedures.

Page 3 Line 128 – “6000 rpm” I recommend converting to “x g”

Section (2.2.2. Hormone treatments) would be more appropriate to present in a table.

Page 4 Line 169 – “centrifuged at 4000 x g” please verify this information, not rpm? Of course, if it is a value in rpm, I recommend converting to x g.

Page 5 Intra and inter assay coefficients are better given in the table, but I will leave this to the decision of the authors.

Page 6 Line 201-205 - The authors report using the CASA system to monitor motility parameters. CASA is a sophisticated system capable of evaluating several sperm quality parameters. Therefore, it is not clear to me why the authors present only three parameters. Specifically, total motility is not as important a parameter as progressive motility. Also, on what basis do the authors present just VCL and LIN? In addition to VCL and LIN, there are other parameters that I consider equally important (DAP – distance average path, DCL – distance curved line, DSL – distance straight line, VAP – velocity average path, VSL – velocity straight line, STR – straightness, ALH – amplitude of lateral head displacement, BCF – beat cross frequency). I am not saying that all of them should have been presented, it is just necessary to explain why the authors present precisely these three parameters of motility in connection with spermiation.

Page 6 Line 217-224 - 2.4. Statistical analysis: Authors claim that the data passed the normality test. I would rather recommend the Tukey´s test than you used in the Total spermatozoon count analyses. Then you need an explanation of why they used Fisher’s LSD test, which is essentially a set of t tests, without any correction for multiple comparisons.

In general, the “Material and methods” section is relatively confusing and extensive. If the authors have already published similar experiments, I recommend a simple citation of previous studies.

  1. Results

There are some ambiguities in this section, but mainly inconsistencies with the previous/next sections. In the “Introduction” and “Discussion” you have a certain structure: Animals → Spermatozoa parameters → Hormonal evaluation. In the “Material and Methods” is this structure half to half. Finally, in the results section you describe the steroids hormones first, followed by sperm quality parameters. Overall, there is a lack of a uniform structure that would increase the clarity and quality of the presented article.

Page 7 line 243-246 In one sentence authors rate the relationship as insignificant. In the next one, results are presented as significant. Again, this part looks like a confusing mix of information. I recommend that you first clearly describe Figure 2a and then proceeding in a straightforward way to second part Figure 2b. Also add the statistical method and test used to the description.

In the 3.2. section is the same problem as the previous one. Describe the results gradually and specifically.

Part 3.3. is basically fine and interesting, however, the addition of other motility parameters would certainly enhance its scientific quality.

  1. Discussion

Although the discussion was merged with the results, it needs several modifications. It should focus on explaining and evaluating what you found (affected spermiation or not?) how it relates to the new research. Extensive adjustment of the discussion will be necessary in case of supplementation/revision of the results.

  1. Conclusion

The authors did not evaluate the possible impact of the studied treatments on spermiations. Finally, it is necessary to explain the fulfillment of the set goals, their contribution, reproducibility, and possible use in the field of reproductive biology. In this form, the conclusion is a strict summary of the results without broader context and associations.

Author Response

We would like to express our gratitude to the reviewer n.3 for the comments leading to overall improvement of the manusctript.

I do not fully identify with the title of the publication. Although the authors monitored some fish spermatozoa quality parameters and hormone production, other complex markers are missing. Spermiation involves a broader complex of observations (for example morphology; ectoplasmic specialization; initiation, progression, and regulation of spermiation etc.) in addition to the monitored parameters. Interactions of various structures, cellular processes and other changes are missing.

Based on the above, I recommend editing the title: e.g. “The effect of hormonal treatment on selected sperm quality parameters and sex steroids in tropical cyprinid bala shark.”

- The recommendation was accepted and the name change accordingly.

  1. Introduction:

The introduction meets the criteria for this type of study, and I have no comments on this section.

  1. Material and Methods:

In this section I have following comments, recommendations, or questions:

Page 2 Line 88 - I recommend adding dimensions of the water tank (the volume itself may not be the authoritative parameter).

- requested information was added

Page 3 Line 128 – “6000 rpm” I recommend converting to “x g”

- requested information was added

Section (2.2.2. Hormone treatments) would be more appropriate to present in a table.

- we agree with this comment and therefore the table was added to the text

Page 4 Line 169 – “centrifuged at 4000 x g” please verify this information, not rpm? Of course, if it is a value in rpm, I recommend converting to x g.

- requested information was added

Page 5 Intra and inter assay coefficients are better given in the table, but I will leave this to the decision of the authors.

Page 6 Line 201-205 - The authors report using the CASA system to monitor motility parameters. CASA is a sophisticated system capable of evaluating several sperm quality parameters. Therefore, it is not clear to me why the authors present only three parameters. Specifically, total motility is not as important a parameter as progressive motility. Also, on what basis do the authors present just VCL and LIN? In addition to VCL and LIN, there are other parameters that I consider equally important (DAP – distance average path, DCL – distance curved line, DSL – distance straight line, VAP – velocity average path, VSL – velocity straight line, STR – straightness, ALH – amplitude of lateral head displacement, BCF – beat cross frequency). I am not saying that all of them should have been presented, it is just necessary to explain why the authors present precisely these three parameters of motility in connection with spermiation.

We do agree with this remark. To select the most informative parameters and for simplifying data presentation we used correlation analysis. Following text was added to manuscript:

“Kinetic parameters obtained by CASA for all sperm samples used in the study were subjected to a correlation analysis using Spearman's rank correlation coefficient. For simplifying data presentation, only parameters with a low correlation coefficient (r < 0.06) were selected as descriptors of sperm motility. These parameters were VCL (curvilinear velocity) and LIN (linearity)”.

Page 6 Line 217-224 - 2.4. Statistical analysis: Authors claim that the data passed the normality test. I would rather recommend the Tukey´s test than you used in the Total spermatozoon count analyses. Then you need an explanation of why they used Fisher’s LSD test, which is essentially a set of t tests, without any correction for multiple comparisons.

Yes, you are correct that Tukey´s test is a good choice here. We used Fisher’s LSD just a potentially valid indicator of possible differences between experimental groups at a particular post-activation time; stable small p (<0.05) for differences between groups (e.g. for sperm motility percent at each time point )can be considered that general conclusion is correct. To avoid uncertainty, we re-analyzed this part of data by Tukey´s test, as you have suggested. We have changed text in Matherial and Methods (L222). The appearance of differences among groups according to the Tukey’s test is described in the corrected version of manuscript:

  1. Results

There are some ambiguities in this section, but mainly inconsistencies with the previous/next sections. In the “Introduction” and “Discussion” you have a certain structure: Animals → Spermatozoa parameters → Hormonal evaluation. In the “Material and Methods” is this structure half to half. Finally, in the results section you describe the steroids hormones first, followed by sperm quality parameters. Overall, there is a lack of a uniform structure that would increase the clarity and quality of the presented article.

Page 7 line 243-246 In one sentence authors rate the relationship as insignificant. In the next one, results are presented as significant. Again, this part looks like a confusing mix of information. I recommend that you first clearly describe Figure 2a and then proceeding in a straightforward way to second part Figure 2b. Also add the statistical method and test used to the description.

- modiefied according to reviewer

In the 3.2. section is the same problem as the previous one. Describe the results gradually and specifically.

- modiefied according to reviewer

Part 3.3. is basically fine and interesting, however, the addition of other motility parameters would certainly enhance its scientific quality.

  1. Discussion

Although the discussion was merged with the results, it needs several modifications.

- according to the reviewer modifications were made in the manuscript

  1. Conclusion

The authors did not evaluate the possible impact of the studied treatments on spermiations. Finally, it is necessary to explain the fulfillment of the set goals, their contribution, reproducibility, and possible use in the field of reproductive biology. In this form, the conclusion is a strict summary of the results without broader context and associations.

- according to reviewer the changes were made

Reviewer 3 Report

Lines 52-56: Please, rewrite this sentence to achieve clarity of the message. 

Lines 57-60: Unfortunate sentence. Please, clarify that you did not mean "extended release of GnRH", but "prolonged or frequent exposure of fish organism to exogenous GnRH". Otherwise you suggest that extended release of GnRH (from its own hypothalamus) leads to elevated LH in blood plasma. Please, rephrase. 

Line 74-77: Own, unpublished observations are not anough for justification of the study. Please, find appropriate justification for the study. 

Lines 40-83: Please, provide state-of-the art knowledge on hormonal stimulation of spermiation in this species. We can get information that it is routine in reproduction of this species, but no other information about the current practices are provided. What are the hormones typically applied, what are the doses of each hormone, temperature and photoperiod effect on hormonal stimulation? All of this is important to understand the novelty of this experiment. 

Line 85: Ethical statement is missing. 

Line 163-164: So, was the lyophilizate diluted in NaCl solution, right? Please, explain here. 

Lines 154-164: There is lack of control group for PLGA - the particles without the GnRH. Also, lack of control group for GnRHa+Met (without Met - how the authors know that Met is needed? The PLGA group suggest that Met is not needed and this could be the cause of poorer performance of GnRH+Met, as it was shown in other fishes Met can be a problem). Additionally, why not the same GnRH form has been used in groups 3 and 4? Besides, why these doses for other hormones were used? There is high species-specificity in terms of dose and effective latency time. Especially, I am interested why such a low dose of hCG was used? This need justification. 

Line 168 and 171: Why 24 h was chosen? According to Fig. 1 the release of the GnRHa is still going well until 72 h. Please, justify the time chosen. Also, how do you know that it was enough time so that the other hormones had chance to play their role and to stimulate the endocrine system alone? 

Line 176: How long the samples were stored prior to motility analysis. 

Lines 196-205: How many spermatozoa was analyzed at each time point? Was it made on some chamber or in a drop? 

Line 243: Maybe too low dose of hCG was used? 

Lines 296-299: Why this sentence stands alone? 

Lines 309-310: How did you presume dopaminergic effect based on the review paper where this species is not described at all? And, when assuming dopaminergic effect, why didn't you test PLGA with Met or other dopamine antagonist? The design of the study need to be more better justified.

Lines 309-314: Any conclusion on that? Was the Met problem? 

Lines 339-341: So, if it was doubtful that the hCG will be used, why it was used? Or maybe the dose was too small? Elaborate on that. 

Lines 367-371: Any advices for the future? 

Author Response

We would like to express our gratitude to the reviewer n.3 for the comments leading to overall improvement of the manusctript.

Lines 57-60: Unfortunate sentence. Please, clarify that you did not mean "extended release of GnRH", but "prolonged or frequent exposure of fish organism to exogenous GnRH". Otherwise you suggest that extended release of GnRH (from its own hypothalamus) leads to elevated LH in blood plasma. Please, rephrase. 

- It was modified according to the reviewer comments

Line 74-77: Own, unpublished observations are not anough for justification of the study. Please, find appropriate justification for the study. 

- Justification of the study is described in lines 48 - 51

Lines 40-83: Please, provide state-of-the art knowledge on hormonal stimulation of spermiation in this species. We can get information that it is routine in reproduction of this species, but no other information about the current practices are provided. What are the hormones typically applied, what are the doses of each hormone, temperature and photoperiod effect on hormonal stimulation? All of this is important to understand the novelty of this experiment. 

- there are no other published or known information about artificial reproduction or reproduction of bala shark than the paper Lipscomb et al (2018) which is already used as reference in our manuscript and dedicated solely to induction of ovulation in bala shark.

Line 85: Ethical statement is missing. 

  • it was placed at the end of manuscript:

    Institutional Review Board Statement: The study was conducted according to the guidelines of the Declaration of Helsinki and followed national and international guidelines for the protection of animal welfare (EU-harmonized Animal Welfare Act of the Czech Republic). The experimental unit is licensed No. 2293/2015-MZE-17214 and No. 55187/2016-MZE-17214 within project NAZV QK1810221 according to the Czech National Directive (the Law against Animal Cruelty, No. 246/1992)."

Line 163-164: So, was the lyophilizate diluted in NaCl solution, right? Please, explain here. 

  • following sentences are part of the manuscript : Male fish were administered a single intramuscular injection ... . All substances were dissolved in 0.9% NaCl.

Lines 154-164: There is lack of control group for PLGA - the particles without the GnRH. Also, lack of control group for GnRHa+Met (without Met - how the authors know that Met is needed? The PLGA group suggest that Met is not needed and this could be the cause of poorer performance of GnRH+Met, as it was shown in other fishes Met can be a problem). Additionally, why not the same GnRH form has been used in groups 3 and 4? Besides, why these doses for other hormones were used? There is high species-specificity in terms of dose and effective latency time. Especially, I am interested why such a low dose of hCG was used? This need justification. 

  • Because of limited number of broodstock we decided to compare PLGA treatment with the most commonly and effectively used hormonal treatment in carp artificial reproduction - Linpe method (combination of GnRHa + dopamine antagonist) and hCG. We are not aware of any information about negative impact of dopamine antagonist addition in case of sperm induction therapies in cyprinids.
  • The same GnRHa was used in the trial.
  • The dose for hCG was used based on standardly used hCG doses in spermiation inducing therapies in freshwater fish with reduced ability to sperm (Zarsky et al., 2020; Cejko et al., 2012 etc).
  • we can exclude the question of latency time because the gonadotropine preparations like hCG act sooner than GnRHa treatments

Line 168 and 171: Why 24 h was chosen? According to Fig. 1 the release of the GnRHa is still going well until 72 h. Please, justify the time chosen. Also, how do you know that it was enough time so that the other hormones had chance to play their role and to stimulate the endocrine system alone? 

- our original plan was to collect sperm at 24h, 48 and 72h post treatment. However due to serious health problems of bala shark broodstock after first collection we had to changed our planes. Simply due to obtain positive results in comparation with other treatments and control, we can conclude that PLGA stimulation was working very well with assumption that later collection would be even more successful (due to long term GnRHa release from PLGA particles).

Line 176: How long the samples were stored prior to motility analysis. 

It is already described in MM: “…stored on ice at 4 °C not longer than two hours during motility analysis

Lines 196-205: How many spermatozoa was analyzed at each time point? Was it made on some chamber or in a drop? 

The total number of spermatozoa in which motility parameters were analyzed at each time point were ranged from 910 to 2774. Altogether 198009 spermatozoa were analyzed (We think these values are not needed in the text of manuscript).

Motility records were made from bottom part of the drop.   

Line 243: Maybe too low dose of hCG was used? 

  • The dose for hCG was used based on standardly used hCG doses in spermiation inducing therapies in freshwater fish with reduced ability to sperm (Zarsky et al., 2020; Cejko et al., 2012 etc).

Lines 296-299: Why this sentence stands alone? 

- it was corrected

Lines 309-310: How did you presume dopaminergic effect based on the review paper where this species is not described at all? And, when assuming dopaminergic effect, why didn't you test PLGA with Met or other dopamine antagonist? The design of the study need to be more better justified.

- we agree with the reviewer and the sentence was deleted

Lines 309-314: Any conclusion on that? Was the Met problem? 

We think that LH profile induced by PLGA treatment was simply more apropriate to stimulate sperm maturation than treatment by GnRHa + Met combination.

Lines 339-341: So, if it was doubtful that the hCG will be used, why it was used? Or maybe the dose was too small? Elaborate on that. 

  • The dose for hCG was used based on standardly and effectively used hCG doses in spermiation inducing therapies in freshwater fish especially cyprinids with reduced ability to sperm (Zarsky et al., 2020; Cejko et al., 2012 etc).

Lines 367-371: Any advices for the future? 

- modified according to rewiever

Round 2

Reviewer 2 Report

Dear authors, 

in my view, you have reworked the article sufficiently.

Best regards.

Reviewer 3 Report

Actually, I am not satisfied with the answers provided. You have not provided strong enough justification of the study. For example, you are stating that hormones are 'routinely applied' in bala shark, and in the answer you are providing information that there is only 1 paper which was a basis for you to claim about 'routine' practice. So, 1 paper is not a routine yet. Besides, I can not accept explanation that you did not create adequate control groups cause you had not enough fish. This is also not acceptable. If you had not enough fish the most scientifically rationale way of designing the experiment is to reduce the experimental groups and provide stronger evidence on the effect of some particular treatment by having appropriate control groups. this consider also the Metoclopramide injected alone - similarly to the lack of evidence in other cyprinids that Met has no negative effect alone, there is also no evidence that hCG has a positive effect in cyprinids (as you claim in your response), or at least you did not provide clear explanation neither in the MS nor in response to review. Only in the response ytou mentioned some papers of Zarsky and Cejko, but they are not cited in the MS so I am not sure what you are referring to. Another point is that the doses of the hormones are not appropriately justified. And despite that you are addressing some of your ideas in response to review there is still lack of these information in the M&M of the MS. So, I consider that my concerns were largely ignored and you intentionally did not want to provide the reader with highly relevant information (they are only in the response to review). So, exept minor corrections you have provided appropriately I sustain my oppinion on the MS: the justification is very weak, the control groups are missing, justification of the doses and latency times are also missing and overal conclusion may therefore considered as biased. Explanation of those aspects (even though I largely disagree with the approach the Authors have undertaken) in response (and not correcting/supplementing the MS itself) only is insufficient, in my view.